# The Influence of Virus Infection on Microglia and Accelerated Brain Aging

**DOI:** 10.3390/cells10071836

**Published:** 2021-07-20

**Authors:** Luis Filgueira, Alexey Larionov, Nils Lannes

**Affiliations:** Anatomy, Faculty of Science and Medicine, University of Fribourg, 1700 Fribourg, Switzerland; alexey.larionov@unifr.ch (A.L.); nils.lannes@unifr.ch (N.L.)

**Keywords:** brain aging, microglia, neuroinflammation, neurotropic virus, HIV, flavivirus, SARS-CoV-2, human herpes virus

## Abstract

Microglia are the resident immune cells of the central nervous system contributing substantially to health and disease. There is increasing evidence that inflammatory microglia may induce or accelerate brain aging, by interfering with physiological repair and remodeling processes. Many viral infections affect the brain and interfere with microglia functions, including human immune deficiency virus, flaviviruses, SARS-CoV-2, influenza, and human herpes viruses. Especially chronic viral infections causing low-grade neuroinflammation may contribute to brain aging. This review elucidates the potential role of various neurotropic viruses in microglia-driven neurocognitive deficiencies and possibly accelerated brain aging.

## 1. Introduction

Aging is a programmed biological process, affecting all biological systems, controlled by genetic [1] and epigenetic mechanisms [2], and influenced by environmental factors [3,4]. The principles of aging apply also to humans [5], to all organ systems and to the brain [6]. Physiological aging of the healthy brain is an age-dependent biological process and consists of deterioration of structure and function [7,8,9,10,11,12]. However, brain aging can be accelerated by multiple factors, due to traumatic events [13,14], following neurovascular conditions [15,16,17], or related to specific brain diseases, including Alzheimer’s and Parkinson’s disease. Accelerated brain aging is often related to enhanced neurodegeneration, which includes loss of neuronal cell structure and function due to (1) metabolic changes [18], (2) neuronal cell death [19] (3) decline in the neuronal network [20], (4) neuronal functional deficiency [21], (5) decline in neuronal regeneration [22,23,24], or (6) a combination of the mentioned reasons. It also includes functional and structural changes of the glial cells, resulting in demyelination and gliosis [25,26,27]. Neurodegeneration is aggravated by neuroinflammation, which contributes substantially to accelerated brain aging. Neuroinflammation usually correlates with the activation of microglia, the resident macrophages and innate immune cells of the brain [28,29]. Thus, the role of microglia in neuroinflammation and brain aging will be explored. Neuroinflammation is often induced by viral infections culminating in encephalitis, which is an inflammatory process of the brain that usually involves the microcirculation, neurons and glia cells, including microglia, as well as infiltration of brain tissue by other cells of the innate and acquired immune system like monocytes, dendritic cells, granulocytes and various subpopulations of T lymphocytes [30,31,32,33]. Encephalitis can be mild, with reversible functional deficiencies, but it can also result in severe structural damage with corresponding functional defects and sequelae [34]. Viruses that affect the brain and may contribute to accelerated brain aging will be reviewed. Finally, the role of microglia in viral brain infections and corresponding accelerated aging of the brain will be unfolded [35].

## 2. Neuroinflammation in Brain Aging

Neuroinflammation relates to a pathological immune response in the brain. It can be sudden and excessive, or subliminal, as well as short-lived or chronic. It may include a cellular immune response of the innate [36] and/or the acquired immune system [37], as well as humoral (antibodies) [38] and soluble factors (chemokines, cytokines) [39,40,41]. It certainly includes general basic cellular responses to inflammatory inducers and mediators (toxins, microbial products, metabolic or other damage-related cellular molecules), recognized by corresponding receptors, i.e., Pattern Recognition Receptors (PRRs) for Pathogen-Associated Molecular Patterns (PAMP) and/or Damage-Associated Molecular Patterns (DAMP) [42,43,44,45,46]. In that respect, all cells of the brain may respond to inflammatory signals, including neurons, astrocytes, oligodendrocytes, microglia, and the cells of the blood vessels (endothelial cells, pericytes, myofibroblasts, vascular dendritic cells, etc.) and the meninges.

There is increasing evidence that the aging immune system is skewed towards a more inflammatory status, increasing the probability and intensity of neuroinflammation [47,48,49]. In addition, a strong systemic inflammatory immune response may influence the brain function and cause corresponding syndromes and disease, as well as induce or aggravate neuroinflammation [50,51,52].

Neuroinflammation interferes with the brain function, may cause structural damage, influence regeneration, and modulate remodeling. It may induce neuronal cell death directly by acting on neurons, or indirectly through actions via astrocytes, oligodendrocytes and microglia, mediated by various neural and inflammatory factors [25,53]. Neuroinflammation and its consequences contribute to physiological brain aging and certainly enhances and accelerates the aging process [19,54]. Neuroinflammation has been shown to contribute to Alzheimer’s and Parkinson’s disease [29,55,56]. It may also play a role in certain psychiatric diseases, including depression, schizophrenia, autism spectrum disorders, etc., some with increasing incidence in aged individuals [57,58,59,60,61].

A plenitude of inflammatory cytokines may contribute to neuroinflammation and its pathological consequences in the brain, including interferons, interleukin-1, 6, 17, 23, and 34, tumor necrosis factor and related cytokines and receptors [62,63,64,65,66,67,68,69,70]. Interestingly, there is evidence that certain cytokines may affect specific brain regions or functional structures. For example, interferon-gamma has been correlated with effects in the hippocampus, where decreased neurogenesis and neuronal apoptosis has been shown in a mouse model [71], whereas interleukin-6 has been shown to disrupt synaptic plasticity [72]. The inflammatory cytokines may enter the brain from the blood circulation through the disrupted blood–brain barrier, or may be produced by infiltrating immune cells, as well as by the local immune cells, i.e., the microglia.

## 3. Contribution of Microglia to Neuroinflammation and Brain Aging

Microglia are the resident macrophages of the central nervous system. They do not belong to the neural lineage like neurons, astroglia and oligodendroglia, but derive from the monocyte/macrophage lineage at an early developmental stage in the yolk sac [73,74]. Microglia precursor cells migrate into the early developing brain, where they contribute to development, homeostasis, structure, and function. However, after birth, bone marrow derived monocytes may immigrate into the brain, where they integrate into the microglia population, being indistinguishable from the original resident microglia [75,76,77,78]. In addition to the parenchymal microglia within the brain tissue, one needs to consider additional macrophage populations inside and around the brain, including perivascular, meningeal and choroid plexus macrophages [79].

The physiological function of resting, non-activated microglia in brain homeostasis is not well understood. Activated microglia may acquire paradoxical, opposite functions, either supporting regeneration and repair, or driving neuroinflammation [73]. The triggers and mechanisms driving the cells towards one or the opposite functional direction are not well understood. However, inflammatory microglia can be harmful and destructive to the brain, whereas regenerative microglia may interfere with physiological brain remodeling processes. Hashemiaghdam and Mroczek (2020) [80] have recently reviewed microglia heterogeneity and known mechanisms for neurodegeneration in the context of Alzheimer’s disease and propose promising research approaches to elucidate it further. For further details and up-to-date information about the role of microglia in brain aging, please see other articles in the corresponding special issue of *Cells* [81,82].

On one hand, microglia contribute to brain homeostasis, cognition and neurogenesis, but on the other hand, they belong to the innate immune system and they also contribute to immune monitoring and control of immune responses in the brain [82,83]. In that respect, they are the tissue resident macrophages and antigen-presenting cells [73]. They respond to local signals related to cell and tissue damage, as well as to microbial invasion of the brain. They have the delicate responsibility of protecting the brain during infection. This includes the control of microbial invasion and elimination of the infectious agent with the help of residential and infiltrating other immune cell populations, including T lymphocytes [84]. T lymphocytes are essential for an efficient and successful immune response against intracellular microbes, including viruses. However, T lymphocytes can be very harmful to the brain, as upon activation they may release high amounts of cytotoxic factors and cytokines, including perforin, granzymes, tumor-necrosis factor alpha, etc., that may damage brain structure and function, but may certainly influence the microglia-immune-network and the microglia function [84,85]. More recently, there is increasing evidence of the presence of tissue resident T lymphocytes in the brain, which may increase in number after corresponding brain infections. These brain resident T lymphocytes may play a crucial role in future brain infections, but also in neuroinflammation and neurodegeneration, and consequently possibly in accelerated brain aging and aging related brain conditions [86,87,88,89,90,91,92,93,94]. However, as microglia contribute to the immune response and resolution of brain infection, they also need to prevent structural and functional damage to the brain cells, to protect the neuronal network, as well as the brain remodeling and neurogenesis process. Consequently, microglia ought to monitor and control the T lymphocytes to make sure that those potent immune cells participate in the elimination of microbes, but without too much harm to the brain. However, it is not well understood how microglia cells manage the balance between the neuroprotective and homeostasis function, and the participation in an inflammatory immune response for the elimination of microbes, which usually also includes the post-infection phase of repair and regeneration.

## 4. Virus Infection in the Brain

Many viruses can infect the brain. Table 1 lists viruses that are of interest in this context and have globally a high incidence, although exact numbers of brain infections are not known. Viruses that target cells of the central nervous system are called neurotropic and may infect neurons, but also other cell types, including glia and endothelial cells. For entering the central nervous system, neurotropic viruses need to overcome the physiological blood–brain barrier [95,96]. Viruses may use a variety of mechanisms to enter the brain, including (1) retrograde axonal transport from the periphery (including the olfactory epithelium in the nasal cavity) as described for herpes viruses [97,98,99], (2) by infecting peripheral immune cells and using them as Trojan horse as described for flaviviruses [95,100] and HIV [101], (3) by infecting brain endothelial cells of the microcirculation and being released toward the basal side [100], or (4) by disrupting the blood–brain barrier and entering directly from the blood circulation [102,103]. Once neurotropic viruses have entered the central nervous system, they need to bind with their surface ligand proteins to corresponding receptors on the target cells enabling them to enter and infect the cells, as reviewed by Schweighardt and Atwood (2001) [104]. However, virus infection of the brain may induce neuroinflammation and neurodegeneration as recently reviewed by Römer (2021) [105]. Certain viruses may specifically target neural or neuronal precursor cells that are needed for neuronal turn-over in certain brain systems, including the hippocampus and the olfactory system [106,107]. Neuroinflammation and neurodegeneration in the course of a viral infection may contribute to brain aging, thus suggesting that viral infections may be a possible etiology of accelerated aging as proposed by Sochocka et al. (2017) [108]. In that respect the study by Lin et al. (2020) [109] needs to be mentioned that indicate that genital human papillomavirus infection may increase the risk of dementia.

Most virus infections cause encephalitis that is usually acute and may be severe, if not lethal. For avoiding acute and severe encephalitis, the virus and its interplay with the immune system should keep the interference with the brain structure and function to a low level [110]. In that respect, one needs to consider that viral infections of the brain may be “asymptomatic” with no short-term effects [111], but viruses may persist in brain cells over a longer period of time [105,110,112]. Accordingly, possible neuroprotective mechanisms for neuronal survival in the course of virus infection may play a role in neuronal persistence of the virus [113]. However, long-term viral persistence may sustain a corresponding immune response and may cause sustained chronic low-grade neuroinflammation that could induce and accelerate the brain aging process [114]. It has also been proposed that frequent and recurrent brain infection with different viruses, with corresponding uninterrupted low-grade neuroinflammation and neurodegeneration, may contribute to accelerate brain aging [115].

Interestingly, many different RNA viruses are neurotropic and may contribute to accelerated aging, including human immunodeficiency virus (HIV), measles virus, several flaviviruses, corona viruses (including SARS-CoV-2), enteroviruses, and influenza virus. However, there are also neurotropic DNA viruses like some human herpes viruses. The following sections explain specificities for each of those viruses, including effects on neuroinflammation, neurodegeneration and cognitive deficiencies, all associated with accelerated brain aging.

Globally, more than 38 million people were living with a HIV infection at the end of 2019, as reported by the World Health Organization [116]. HIV infects microglia and under certain conditions also astrocytes, with excessive neuroinflammation, and neuronal and glial cell death when untreated, resulting in loss of brain structure and function [117,118,119,120,121]. Using post-mortem analysis of patients with chronic HIV infection, Taber et al. (2016) reported that HIV infection resulted in activation of microglia cells and astrocytosis in the hippocampus and the neocortex, indicating HIV-related neuroinflammation which possibly induced decreased neurogenesis, as well as neuronal cell death [118]. Even under antiretroviral therapy (ART), decline in neurocognitive functions, similar to aging, has been described due to persistent HIV brain infections, and named HIV-associated neurocognitive disorders (HAND) [112,122,123]. The detailed cellular and molecular mechanisms responsible for HAND have still to be elucidated [124,125]. However, chronic HIV infection results in corresponding neuroinflammation with activation of microglia and astrocytes [126,127,128]. There is also increasing evidence that there is production and deposition of beta-amyloid peptide, similar to Alzheimer’s disease [129,130]. Furthermore, oligodendrocytes are affected resulting in their dysfunction, demyelination and white matter loss [131].

The global infection rate with measles virus has substantially decreased due to a successful vaccination program, as reported by the WHO in 2020. However, there were still 207,500 measles deaths in 2019, indicating the importance of measles virus infection at the global level [132]. Measles virus infects neurons, as well as oligodendrocytes, astrocytes and microglia and it can spread from one cell to another [133,134,135]. Measles virus may persist in the brain, as shown through animal experiments [136,137,138]. Measles virus infects mostly children or younger adults and causes usually acute encephalitis, but may occasionally induce subacute sclerosing panencephalitis (SSPE) [139]. SSPE is caused by persistent infection with a defective measles virus and includes cognitive decline, and other symptoms and syndromes that may be seen in advanced accelerated brain aging [140,141,142]. SSPE correlates with chronic neuroinflammation and results in neuronal loss and demyelination, typical events in accelerated brain aging.

Many flaviviruses are neurotropic, including Japanese encephalitis virus (JEV), Zika virus (ZIKV) and tick-born encephalitis virus (TBEV) [100,143,144]. It is estimated that there are about 400 million newly infected people annually with flaviviruses in all geographic regions of the globe [145]. They may infect all different cell types of the central nervous system, including neurons, astrocytes, microglia and endothelial cells [100,143,146,147,148,149,150]. Often, when entering the brain, these viruses cause acute encephalitis. Nevertheless, they can persist in the brain [151,152,153,154]. However, there is paucity of knowledge about chronic effects of persisting virus in the brain and corresponding neuroinflammation. Interestingly, JEV can persist in microglia without production of infectious virus, but still activate the cells and possibly sustain chronic neuroinflammation [151,155]. The same accounts for TBEV that may persist in astrocytes [147,154,156], as well as for ZIKV that may persist in brain endothelial cells [152].

To date, SARS-CoV-2 is of special interest due to the present pandemic [157,158], as more than 180 million cases of confirmed infected people have been reported by the World Health Organisation since the beginning of 2020. This corona virus is able to infect brain cells, including neurons, astrocytes and microglia [159,160,161]. It is not known whether the virus can persist in the brain. However, besides brain infection and corresponding encephalitis, the virus may induce a strong systemic inflammatory reaction through a cytokine storm, which includes production of interleukin-17, which substantially affects brain functions and may result in structural damage [159]. As SARS-CoV-2 is new to the human population, it is difficult to tell, whether it will have an effect on accelerating brain aging. However, longer term effects and residual symptoms and syndromes, called COVID-19 encephalopathy, have been recorded in many patients of various ages, including psychiatric effects and cognitive deficiencies [160,162,163,164,165,166,167,168]. Interestingly, experimental depletion of microglia in a mouse model indicate that microglia may protect from SARS-CoV-2 related brain damage [169].

Similar information to what has been said about the viruses above, accounts also for other RNA viruses, including various enteroviruses [170,171,172,173,174,175] and influenza viruses [106,176,177,178].

Finally, human herpes viruses, including herpes simplex (type 1, 2, 6), varicella Zoster and Epstein–Barr virus, which are DNA viruses, have also neurotropic properties, may infect the brain and induce chronic neuroinflammation and neurodegeneration [179,180,181,182]. There may be an association between brain infection with herpes viruses and accelerated aging, including dementia and Alzheimer’s disease [183,184,185,186,187,188,189,190,191]. In addition, there is evidence that microglia may play a protective role against pathological effects of herpes viruses [192]. Although no exact data about the incidence of herpes virus infections is known, the World Health Organization assumes that the majority of the global human population, i.e., several billion people, suffer from a lifelong infection [193]. With this in mind, brain infection with herpes virus may also be quite frequent, although exact numbers are not available.

Summarizing, there is increasing evidence that microbial agents, and especially viruses, that infect the brain may induce chronic neuroinflammation, which may contribute to neurodegeneration and accelerated aging [194,195]. The detailed mechanisms of infection-related chronic neuroinflammation still need to be elucidated and they may depend on the specific infectious agent and the corresponding immune response. However, microglia in their role as innate immune cells of the brain ought to have a substantial contribution in pathological processes by losing control of the physiological resolution of the infection. The microbes themselves and the corresponding immune response for the control and elimination of the infection, as well as the subsequent repair mechanisms may individually or altogether act on the microglia. By doing so, these events may cause microglia dysfunction and interfere with microglial essential neuroprotective functions, needed for brain homeostasis, remodeling and neurogenesis. Microglia dysfunction, initiated by acute or chronic brain infections may also contribute to neurodegeneration with increased neuronal cell death, demyelination and disruption of the neuronal network resulting in accelerated aging, as well as contributing to corresponding brain conditions like Alzheimer’s disease [196].

## 5. The Role of Microglia in Viral Brain Infections Accelerating Brain Aging

Microglia are essential to the healthy brain, as they contribute to many brain functions and help sustain the physiological brain structure [83,214,215]. Belonging to the innate immune system of the brain, microglia contribute to an immune response against any brain-invading agent [35,216,217], as well as following traumatic and neurovascular brain damage [218,219,220]. They help with resolving tissue damage and support regeneration and restauration of structure and function [74]. However, microglia may get out of control and out of balance resulting in augmenting brain damage or sustaining chronic pathologies like neuroinflammation and inducing or enhancing neurodegeneration [28,35]. In such case, microglia may enhance brain aging [59,80,128,196,221,222,223,224,225,226].

As part of the immune system, microglia are the sentinel cells of the brain and sensors for invading infectious agents [42,227]. In the context of viral infections, microglia orchestrate the local immune response [35,217,228]. They may help with recruitment of blood-derived immune cells, including monocytes and T-lymphocytes, for fighting and eventually eliminating the infectious agent. Often, this process appears in the form of encephalitis, an immune response with a strong inflammatory component. Encephalitis can be mild, but it can also result in substantial brain tissue damage with either long-term functional deficiencies or even death. However, microglia are probably the master controllers of the process, ensuring on one hand the elimination of the virus and on the other hand preventing damage and promoting repair.

Various infection-related factors can interfere with microglia functions as master controllers. Firstly, microglia can be infected with the invading virus, as it has been shown for HIV, JEV, ZIKV and SARS-CoV-2 [117,148,229,230], which may undermine the microglia support of fighting the infection, as well as reduce their repair and regeneration capacity. Secondly, microglia may respond with excessive production of neurotoxic factors including oxygen radicals (ROS) [231]. ROS production by microglia has been reported in the context of brain infection with various viruses [127,232,233,234,235]. ROS has been shown to contribute to neuroinflammation, neuronal cell death and corresponding neurodegeneration, thus enhancing brain aging [236,237,238]. Consequently, viral infections of the brain may well contribute to accelerated brain aging by inducing a neurotoxic immune response in microglia. Thirdly, the cytokine response related to the elimination and resolution of the infection may be out of control with inflammatory cytokines dominating, causing damage and preventing repair and regeneration, as well as interfering with physiological brain functions. Such damaging cytokines are interleukin-17 [63,64,65,159,239,240] and tumor necrosis factor [39,67,241], whereas interferons and interleukin-6 seem rather to support viral clearance and repair [62,192,242,243]. Inflammatory cytokine may not only be produced locally in the brain, but the infection may induce excessive production of cytokines in the periphery with corresponding release into the blood circulation, which may substantially influence the blood–brain barrier and the entire brain [50]. The various virus infections mentioned above can induce systemic production of excessive amounts of inflammatory cytokines, e.g., called cytokine storm in the context of the SARS-CoV-2 infection [244]. Fourthly, viruses may remain for a long time in the brain and sustain a chronic low-grade inflammatory response, as already mentioned in Section 4. All these factors can affect microglia function in the long term, resulting in sustained low-grade inflammatory or dysfunctional microglia with decreased support of brain regeneration and remodeling.

## 6. Conclusions

The more we age, the more our immune system gets toward a more inflammatory status. The increased systemic inflammatory immune status also affects microglia, resulting in decreased physiological neuroregeneration and remodeling [49,245]. The inflammatory status is certainly enhanced and accelerated through frequent or chronic viral infections. The increased and chronic inflammatory status in the brain may contribute to neurodegeneration due to increased neuronal cell death and reduced neurogenesis, reduced remodeling and irreparable damage to the neuronal network, resulting in an enhanced or accelerated brain aging process. In the context of microglia and viral infection, most research has been done in HIV, where the association has been shown for neurocognitive decline [127,128,228,246]. However, there is little information available about the cellular and molecular mechanisms that contribute to or influence the chronic HIV infection and corresponding involvement of microglia, which requires more future research. The same accounts for other viruses, including flaviviruses, human herpes viruses and SARS-CoV-2.

## Figures and Tables

**Table 1 cells-10-01836-t001:** Neurotropic viruses, target cells, scientific and experimental evidence for persistence in the brain, effects on the brain and clinical appearance ^1^.

Virus		Infected Cells	Scientific and Experimental Evidence for Persistence in the Brain [Ref.]	Effects on the Brain	Clinical Appearance
Retrovirus (single-stranded positive-sense RNA virus)	HIV [112,117,124,197,198,199,200,201]	Microglia, endothelial cells, astrocytes	Clinical studies, diagnostics of cerebrospinal fluid, histopathology [111,112,199,200,201]	Neuroinflammation, break-down of blood-brain barrier,	HIV-associated neurocognitive disorders (HAND)
Paramyxovirus (single-stranded negative-sense RNA virus)	Measles virus [135,202,203]	Neurons, astrocytes, microglia, oligodendrocytes	Experimental animal studies, clinical observation [110,135,202,203]	Neuroinflammation, neurotoxicity	Encephalitis, subacute sclerosing panencephalitis
Flaviviruses (single-stranded positive-sense RNA virus)	JEV [100,151,153,155]	Microglia, endothelial cells, neurons, astrocytes	In vitro cell culture studies [100,151,155]	Neuroinflammation, break-down of blood-brain barrier, neurotoxicity	Encephalitis
	ZIKV [143,146,148]	Endothelial cells, neurons, microglia	Experimental animal studies [143,153]	Break-down of blood-brain barrier, neurotoxicity, neuroinflammation	Meningoencephalitis
	TBEV [147,149,150,154]	Astrocytes, neurons, endothelial cells	In vitro cell culture study	Neuroinflammation, neurotoxicity, virus transfer from blood to brain tissue	Encephalitis
Corona (single-stranded positive-sense RNA virus)	SARS-CoV-2 [159,160,161]	Neurons, astrocytes, microglia, endothelial cells	In vitro cell culture study, experimental animal study, post-mortem histopathology [161]	Neurotoxicity, neuroinflammation, break-down of blood-brain barrier	Cognitive and Neuropsychiatric manifestations
Enterovirus (single-stranded positive-sense RNA virus) [170,172,173,204,205].	Non-polio enterovirus	Neurons, endothelial cells, astrocytes	In vitro cell culture studies [173]	Neurotoxicity, neuroinflammation, break-down of blood-brain barrier	Encephalitis
Influenza (single-stranded negative sense RNA virus) [176,177,178,206,207].		Neurons, astrocytes, endothelial cells	Experimental animal studies [176,177]	Neurotoxicity, neuroinflammation, break-down of blood-brain barrier	Encephalitis
Herpes (double-stranded DNA virus)	Herpes simplex virus (1, 2, 6), Varicella Zoster virus, Epstein-Barr virus [179,187,188,189,208,209,210,211,212,213]	Neurons, astrocytes, endothe lial cells	Experimental animal studies, clinical observation [180,183,210,211,212]	Neurotoxicity, neuroinflammation, break-down of blood-brain barrier	Encephalitis, Dementia?

^1^ References: Immune deficiency virus (HIV): [112,117,124,197,198,199,200,201]. Measles virus: [135,202,203]. Japanese encephalitis virus (JEV): [100,151,153,155]. Zika virus (ZIKV): [143,146,148]. Tick borne encephalitis virus (TBEV): [147,149,150,154]. Severe acute respiratory syndrome coronavirus 2 (SARS-CoV-2): [159,160,161]. Enterovirus: [170,172,173,204,205]. Influenza virus: [176,177,178,206,207]. Human herpes viruses: [179,187,188,189,208,209,210,211,212,213].

## Data Availability

Not applicable.

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
