# Peer review of "The Influence of Virus Infection on Microglia and Accelerated Brain Aging"

_cells, 2021, doi:10.3390/cells10071836_

Round 1
Reviewer 1 Report
In this review entitled “BRAIN AGING INFLUENCED BY VIRUS INFECTION AND MICROGLIA” by Filgueira, Larionov, and Lannes, the authors have aimed at highlighting the potential interdependencies between viral infections of the brain, microglia-mediated neural inflammation, and accelerated brain aging. I believe this review to be relevant and timely, particularly with regards to assessing viral infections as a potential driver of brain aging and the specific heed paid to the SARS-COVID-19. Overall, the manuscript is presented in a clear and systematic manner and covers sufficient literature to realize the aim of the review. However, there are some areas for improvement, as detailed below. These changes mainly pertain to an over-interpretation of literature. The following revisions are suggested, mainly to aid in strengthening the presentation of the key outcomes of the review prior to publication:
Major comments:
- Title: There is a discrepancy between the title of the review and the contents therein. Therefore, it is suggested that an alternative title is utilized, one that more accurately represents the main ideas of the review.
- In Table 1 [line 133-134], and a later paragraph [175-186], the authors claim that "This corona virus is able to infect brain cells, including neurons, astrocytes and microglia". I know some of these publications in detail, and upon further assessment and evaluation of the literature, the conclusion drawn here appears to be not only over-simplified, but in fact incorrect. References 108 and 109 do not provide sufficient evidence to support this textual shortage. The narrative of references 108 and 109 only allow for in-vitro based conclusions with excess virus titers. It seems that in the field, these studies do not provide robust evidence that supports the infection of brain cells in a realistic context of virus encounter, and were rather published in a hurry with somewhat alarmistic intentions. The exception are the data regarding endothelial and other adjacent non-neural lineage cells that have been more solidly demonstrated to be infected by SARS-CoV-2 in e.g. post-mortem patient samples. It is therefore suggested that the authors reformat Table 1 to provide a more detailed insight as well as allow for direct scrutiny of the robustness and validity of the data. This can be achieved by displaying both the validation method behind each claim as well as the corresponding reference as additional columns in Table 1. Further, blank fields should be avoided, or text centred in the middle of the field. For example, in row one column six – it’s not clear whether HIV-associated neurocognitive disorders is only associated with microglial infection or with endothelial and astrocytic infection as well. The same accounts for the persistence column.
Minor comments:
- While the manuscript is coherent, many grammatical errors are present within the text. It is therefore suggested that this manuscript is proofread prior to publication. E.g. lines 115-116 features an incorrect citation.
- In lines 18-19, the distinction of “and most animal species” should be left out.
- Line 27: Missing “3)”
- Line 29-30: Please supply a reference for “It also includes functional and structural changes of the glial cells, resulting in demyelination and gliosis.”
- In lines 49-50, “and the cells of blood supply” should be clarified.
- Line 69: Please expand on “effects in the hippocampus”
- Line 145, please elaborate upon the “loss of structure and function” of microglia due to HIV infection
- The manuscript presents the idea of neuroinflammatory stimuli as a potential driver for brain aging at several points: Line 138 “accelerated aging”, Line 147 “similar to aging”, Line 182 “accelerating brain aging”, Line 192 “accelerated aging”. However, in line 236 aging is presented as a cause of inflammation – not as a consequence of it which counteracts the main idea of the manuscript. Overall, the link between an inflammatory and aged phenotype should be established – therefore literature regarding a systematic comparison, assessing phenotypic changes in both scenarios, should be reviewed.
- Additionally, I would ask the authors to provide a more detailed view with regards to the local acute or chronic microglia-mediated immune response [Lines 206-234] in response to a viral infection considering specific pathogens. This information could be woven into earlier paragraphs in section four “[Viral] infections [of] the brain”
- Line 177 regarding SARS-CoV-2: “It is not known whether the virus can persist in the brain”. This is in contrast with Table 1, where there is a “yes” under the persistence column for SARS-CoV-2.
- Line 185-186: Please specify that this was performed through the use of an animal model to avoid misinterpretation
- Line 211, Encephalitis has been mentioned throughout the manuscript but is only described here, please define encephalitis earlier in the manuscript.
Author Response
Response to Reviewer 1:
We appreciate very much the feedback and suggestions of reviewer 1 with our answer as follows:
- We changed the title, which says now: ¨The Influence of Virus Infection on Microglia and Accelerated Brain Aging¨
- We fully agree that there is not yet much sound scientific evidence in the context of brain infection by SARS-CoV-2. However, one has to consider that this emerging virus has only recently affected the human population. Nevertheless, there is an increasing number of studies published about brain infection with this coronavirus. We added now a new reference by Eric Song et al., 2021, DOI:10.1094/jem.20202135, that provides supporting data for infection of human neurons by SARS-CoV-2 using in vitro human, in vivo humanized mouse and post-mortem human models.
We tried to optimize the format of Table 1. However, it is difficult to do so in the text and we leave it to the editorial office to do the formatting. If required, we can certainly provide the original Excel table.
Minor Comments:
- The text has been thoroughly proofread and hopefully there are no remaining spelling mistakes. Otherwise, if required, we would appreciate to have the support by the Journal Editorial Office and pay for that service.
- The statement ¨and most animal species¨ has been deleted, as suggested.
- The mistake has been corrected as suggested.
- We added 3 references to the statement about ¨glial cells, resulting in demyelination and gliosis¨
- We replaced ¨blood supply¨ by ¨blood vessels¨ and added the corresponding cells.
- The effects on the hippocampus have been briefly specified.
- ¨loss of structure and function¨ was rather in the context of the brain and not related to the cellular level, which has been now specified in the text. However, one specific sentence related to microglia activation and astrocytosis in brains of HIV patients has been added.
- It was not our intention to say that brain aging as such causes neuroinflammation. Aging in that context is related to the aging of the immune system and systemic immune responses. In that sense, microglia representing the brain immune cells ought also to be directly or indirectly affected by the aging process of the immune system. It was not intended to say that the brain aging process as such affects microglia and related neuroinflammation. To prevent confusion, the first sentence of the Conclusion paragraph has been modified and a second sentence added to improve the statement and to make it better understandable. A corresponding reference has been added. However, whether microglia are directly affected by the brain aging process itself and drives them toward more inflammatory functions is certainly a tempting speculation, remains to be proven and may be discussed in another context.
- We included in Section 3 an additional paragraph about the role of microglia as antigen-presenting cells of the brain that contribute to the control and resolution of brain infections, and in that respect that microglia may control corresponding T lymphocyte function. We added brief information with corresponding references that microglia and T lymphocytes form an immune network that has not been well studied, to date, but which may play an essential role in brain homeostasis an pathologies, including accelerated brain aging. This paragraph covers the immune response against brain invading pathogens in general.
- Table 1 has been adjusted and ¨yes¨ has been replaces by a ¨?¨ for SARS-CoV-2.
- The information has been added that the experimental work of the reference had been done in a mouse model.
- Additional information about encephalitis is now included in the introduction.
Reviewer 2 Report
Title: Brain Aging Influenced by Virus Infection and Microglia The manuscript is a review on the potential role of various neurotropic viruses in microglia-driven neurocognitive deficiencies and possibly brain aging. The following is my opinion and all of them are minor.
Minor points
1.Brain aging and microglia almost automatically bring neurodegeneration issue.Although the authors briefly mentioned them in line 25, 62, 92 and 193, I think moredescription of connection between neurodegenerative disease and viral infection willbe helpful.
2.Related to 1, oxidative stress is one of major causes of neurodegeneration. Is viralinfection directly related to oxidative stress? If so, describe it in the text.
3.It will be informative if the annual numbers of such viral infections in Table 1 that hasbeen reported are indicated.
4.Please add classification of virus based on its genome (whether they are RNA or DNAvirus) in Table 1.
Author Response
Response to Reviewer 2:
We thank reviewer 2 for the recommendations that have been integrated accordingly in the manuscript as stated below:
- We have added information about the relationship between viral brain infections and neurodegenerative disease at the end of section 4.
- We have added information about the role of ROS in in neurodegeneration with corresponding additional references in section 5.
- We have now included information about incidence of the various viruses in the text of section 4, but not in Table 1.
- We include now in Table 1 the information about whether they are RNA or DNA virus
Round 2
Reviewer 1 Report
In the revised manuscript now entitled “THE INFLUENCE OF VIRUS INFECTION ON MICROGLIA AND ACCELERATED BRAIN AGING”, the authors highlight the microglial involvement in viral infections of the brain. They included additional information regarding the contribution of microglia in the specific context of a pro-inflammatory environment that mimics that of an aged brain.
However, I still feel that additional information and slight restructuring of the information presented seems necessary to increase the usefulness of this article and to recommend this manuscript for publication in Cells. It seems that some of the primary articles have not been thoroughly summarized in some cases, and a critical evaluation of the primary data in the scientific context is lacking.
Comments:
- For some of the data presented the correct references are not indicated. e.g. Line 151, 164: Epidemiology data by the World Health Organization (WHO)
- Please indicate the source publication and the concrete method use to show a pathology-relevant implication: e.g. Line 153: Microglia-Involvement under certain conditions => Please elaborate this claim. Which are the conditions that trigger a microglia response, what does this result in for neural viability and functional mechanistic?
- To avoid misinterpretation of insufficiently/naively summarized data, I suggest that at least Table 1 should be revised, with an additional column stating the method of validation. This could state e.g. in-vitro assay (IVA), post-mortem data (PMD), etc. for the respective studies. Here it should be marked where the current state of research does not allow a conclusive statement. In case formatting restrictions do not allow for a different layout of Table 1, I feel that a more critical discussion of some of the ‘catchy’ claims should be provided. E.g. Line 193-200 relating to Herpes simplex virus “may infect the brain and induce chronic neuroinflammation” Which cell types are involved? Is the virus (HSV1/2, HHV-3, EBV) persistent in these cells and which clinical implication arise thereby? How were these observations established?
- Perhaps the addition of a figure or a graphical abstract representing the molecular mechanism of each virus infection on neurons and especially microglia would be beneficial.
- Proofreading and sentence structure should be revised. E.g. odd wording Line 133/134:
In that respect the study by Lin (2020) et al. [109] has also to be mentioned that indicated that…
In that respect a study by Lin et al. (2020) [109] needs to be mentioned, which indicates that...
- The authors should ensure that the additional information is not redundant with the information already present within the text.
Author Response
Second Response to Reviewer 1:
We appreciate very much the new feedback and suggestions of reviewer 1 with our answer as follows:
- We added the missing references for HIV, Measles virus and Herpes virus
- Unfortunately, the line numbers change and are different when using different computer systems for MS Word, as the text is immediately newly formatted when the same file is opened. Therefore, it is not clear in what context the role of microglia in neuro-pathologies may be extended and more details provided. We could certainly add more corresponding information throughout the text. However, the authors feel that the included references provide enough information about a specific topic if the reader wants to go deeper. In addition, one could compile another separate manuscript, just about the influence of microglia biology and mechanisms on neuro-pathologies. In the context of viral infections of the brain, there is not much known, especially in humans. Also, each different virus has its own different effects on microglia that probably respond correspondingly in different ways. We feel that more research is needed for all those viruses to better understand what really is going on in viral brain infections and how they may influence microglia and subsequently all various neuro-pathological events.
- Table 1 has been changed and adapted according to the recommendations, and additional references added, supporting the provided information.
- As the topic is complex, it is difficult to provide a graphical abstract, or helpful schematic drawings that cover the key points of the manuscript. However, if required, we will provide those, but we would need substantially more time.
- Thank you for the specific recommendations of what to change, which has been done in the revised text. We have carefully proof-read and checked the text before resubmitting.
- We hope that there is enough redundancy in the text to emphasize on the relevant and important issues of the topic, but the text still being concise and interesting for the reader.